# A positive statistical benchmark to assess network agreement

**Bingjie Hao** [1] **& István A. Kovács** [1,2] ✉

Current computational methods for validating experimental network datasets compare overlap, i.e., shared links, with a reference network using a negative benchmark. However, this fails to quantify the level of agreement between the two networks. To address this, we propose a positive statistical benchmark to determine the maximum possible overlap between networks. Our approach can efficiently generate this benchmark in a maximum entropy framework and provides a way to assess whether the observed overlap is significantly different from the best-case scenario. We introduce a normalized overlap score, Normlap, to enhance comparisons between experimental networks. As an application, we compare molecular and functional networks, resulting in an agreement network of human as well as yeast network datasets. The Normlap score can improve the comparison between experimental networks by providing a computational alternative to network thresholding and validation.

We are witnessing a rapid expansion of network data, transforming a broad range of scientific fields[1]. In network representations, entities of interest are represented by nodes and connected by links. The degree of each node is given by the number of links it participates in. Real-life networks often have a broad degree distribution, marked by the presence of highly connected hubs[2]. In social and biological applications, multiple network maps exist that capture different modalities of relationships between the same nodes, often leading to limited overlap. As a key example, various biophysical relationships between proteins are critical to biological processes in cells. There are two major categories of the corresponding experimental assays, binary methods, such as Yeast-Two-Hybrid (Y2H)[3–7], and non-binary methods, such as affinity purification followed by mass-spectrometry (AP-MS)[8–11]. Such biophysical measurements have provided a wealth of overlapping yet distinct experimental data about both binary and non-binary protein–protein interactions (PPIs). Note that, even with recent advances in biotechnology, it remains fundamentally impossible to obtain complete maps of macro-molecular networks using only a single assay[12], limiting the overlap between the observed networks. In addition, functional networks such as genetic interaction (GI) networks[13,14] identify functional relationships among the genes or their corresponding gene products. Assessing the agreement between networks within or across different modalities is a valuable step toward utilizing any of these networks for a comprehensive understanding of the underlying biological processes.

Until now, reliably assessing the agreement of different networks remained elusive. Networks are routinely compared by either computationally counting the overlap (shared links) with a suitable reference network or experimentally with a complementary assay[3,4,12,15]. However, such a comparison often leads to a disturbingly low overlap, even between networks of the same modality[12], due to differences in the search space, as well as assay and sampling limitations[15]. For example, in the systematic genome-wide human binary protein interactome (HuRI)[4], among 52,569 PPIs observed in the combination of three experimental Y2H assays, only 2707 (5%) PPIs are supported by at least two assays, while only 258 (0.5%) are found in all three assays. In this specific example, the assays are complementary by design, rendering a direct interpretation of such a low overlap misleading. However, in comparisons where the context of the dataset is less clear, a direct interpretation of low overlap naturally raises the question: Is low overlap a result of low network quality or biophysical differences?

The answer may be neither. Previous studies have shown that even if all measured interactions are true positives, the low observed overlap could stem from multiple sources[15]. First of all, current biological networks are still highly incomplete[4,5,16]. Taking the most studied organism–yeast–as an example, a combination of PPIs from four

[1]Department of Physics and Astronomy, Northwestern University, Evanston, IL 60208, USA. [2]Northwestern Institute on Complex Systems, Northwestern University, Evanston, IL 60208, USA. ✉e-mail: istvan.kovacs@northwestern.edu

systematic Y2H assays covered only 12–25% of the estimated yeast binary interactome[5]. Hence, a low overlap could arise from the high level of incompleteness, especially if the data coverage is more homogeneous in some datasets than in others[17]. As in the case of HuRI, experimental assays are often designed to be complementary to each other to unveil as much new information as possible[12]. Furthermore, different assays will preferably capture some interactions versus others[5,12,17], leading to the detection of distinct sets of links and, thus, low overlap between the various datasets. As a result of these factors, the observed node degrees can be inconsistent across network maps,

even if the degree distributions appear to be similar. For example, in Fig. 1a, b, we illustrate that a hub in one network may have a small degree in the other network and vice versa, causing the degree correlation plot to be off-diagonal, inherently leading to a small overlap. The problem goes beyond the extent to which the reported node degrees can be trusted. Such node degree inconsistency questions the validity of using the observed overlap as a measure of network agreement since a low overlap could either originate from a degree inconsistency or other factors like quality issues and biophysical differences. The use of overlap to assess network agreement is especially

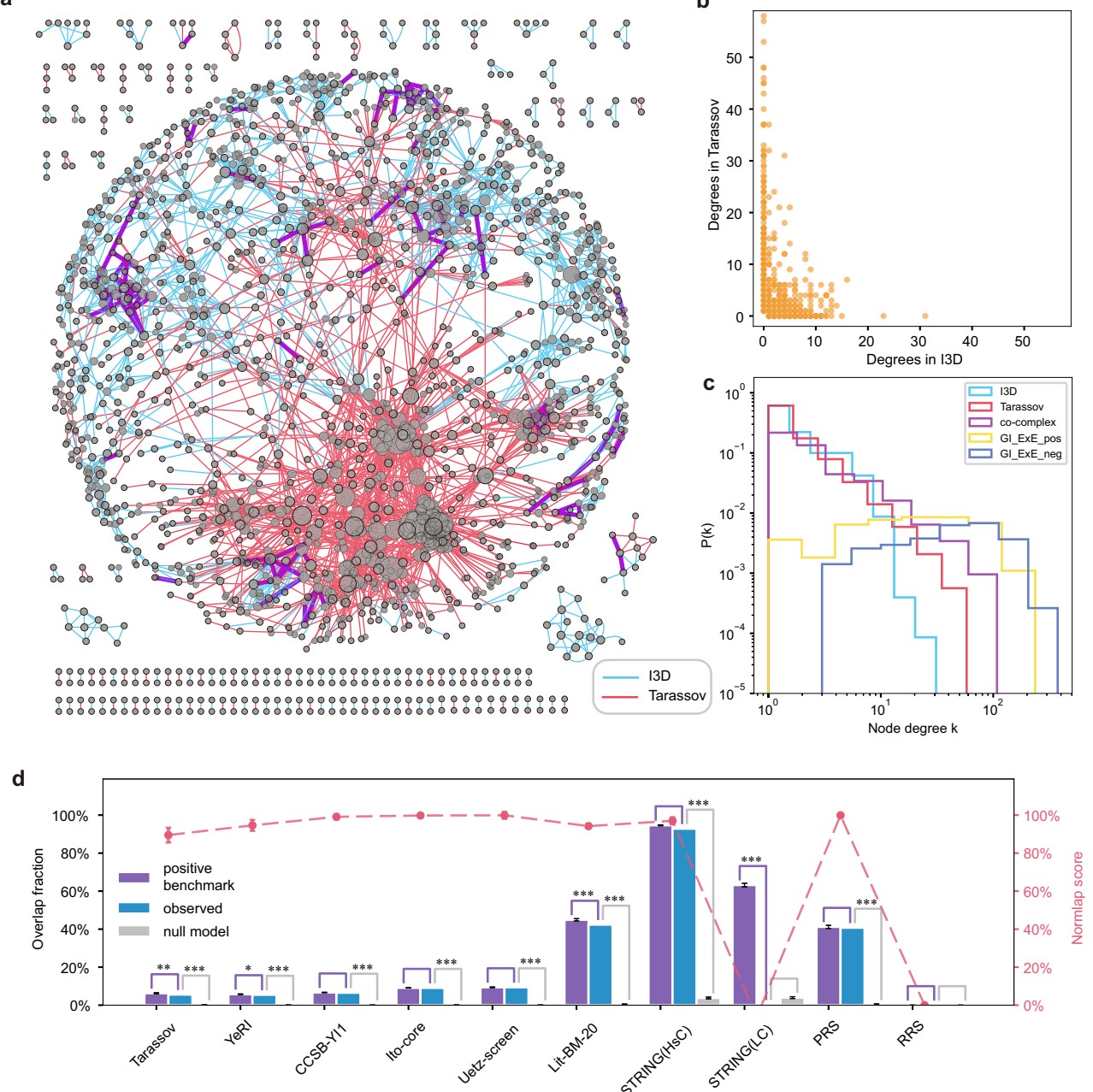

**Fig. 1 | Yeast PPI networks compared to the reference network I3D[23] and the null model. a** The network structure of I3D (blue) and Tarassov[11] (red). The 137 overlapping links (2%) are colored in purple. **b** Degree inconsistency between I3D and Tarassov networks. **c** Node degree distribution for various yeast networks. **d** Overlap fraction of various PPI networks compared to I3D. The proposed positive benchmark is in purple, leading to the normalized overlap (Normlap) score in red. In gray, we show a degree-preserved randomized I3D as a null model for reference.

The significance of the observed overlap compared to the null model or the positive benchmark is determined by a one-sided $p$-value (see "Methods") and indicated by *$p < 0.05$, **$p < 0.005$, ***$p < 0.0005$. Note that the benchmark, Normlap score, and related standard deviation (SD) are directly calculated instead of averaged across random samples, as described in "Methods". Data are presented as mean ± SD.

problematic when comparing different types of networks, where the degree distributions could be very different. The example Fig. 1c shows that binary physical networks showcase a broad degree distribution, while functional networks such as GI networks tend to have a more narrow distribution with no major hubs. Thus, node degrees need to be considered as confounding factors in network comparison. This is routinely achieved by state-of-the-art methods to generate negative benchmarks (null models). In such a negative benchmark, network randomization is performed while preserving node degrees either exactly or on average to compare network characteristics[18,19]. Although a significant difference from the negative benchmark indicates the presence of signal in the network of interest, it fails to quantify the amount of signal.

Here, as the so far missing positive side of the assessment, we set out to establish a positive statistical benchmark to compare against. Such a positive benchmark must be a generative model that preserves node degrees so that any observed variable in a given network can be compared against the benchmark ensemble. In the proposed positive benchmark, every link in both networks is assumed to be sampled from the same underlying network of real connections, according to the observed degrees. Such a positive benchmark corresponds to a best-case scenario, where the two networks are the same as far as the node degrees allow. The overlap between the two sampled networks in the best-case scenario is regarded as the positive benchmark for network overlap. With both the negative and positive benchmarks at hand, we can then place the observed overlap on a scale between 0 (null model) and 100% (best-case scenario), leading to the normalized network overlap or Normlap score.

Owing to the availability of a broad range of molecular and functional network datasets in yeast, we selected *S. cerevisiae* as the primary model system, in addition to a smaller cohort of human networks, as listed in "Methods". In both organisms, we create an agreement network of networks, illustrating the knowledge landscape of the available datasets. In addition, we illustrate how the Normlap score can provide a computational alternative to validate experimental network maps[20,21].

## Results

### Studying a negative benchmark and the impact of degree inconsistency

As the current best practice, the negative benchmark is obtained by taking the overlap between the network of interest and a degree-preserved randomized reference network. Here, we construct a negative benchmark using a maximum entropy framework[19,22] (see "Methods" for details). As a starting point, we compare the observed overlap of 10 yeast PPI networks with a reference network and its randomized version as a null model in Fig. 1d. As our first reference network, we select the I3D[23] network since all of its PPIs are supported by structural evidence. Note that I3D has well-understood study biases due to the fact that some proteins are more easily crystallized than others, leading to higher node degrees. Later we will also consider other datasets as reference networks (Supplementary Fig. S2), although there is no dataset without limitations. In the following, for simplicity, we say that there is network signal when the observed overlap is significantly higher ($p < 0.05$, one-sided $p$-value, see "Methods") than the negative benchmark. All studied networks show network signal when compared to I3D except for the random reference dataset (RRS)[3] and a low-confidence (LC) PPI network compiled from the STRING database[24], as shown in Fig. 1d. Another observation is that all systematic networks (Tarassov[11], YeRI[5], CCSB-YI1[3], Ito-core[6], and Uetz-screen[5]) have significantly lower overlap (<10%) than the literature-curated network Lit-BM-20[5] and the highest confidence (HsC) network from STRING[24], and the positive reference dataset PRS[3]. This observation is in line with the more homogeneous coverage of the interactome by systematic networks compared to integrated

networks, as shown previously in ref. 5. As an underlying reason for the low overlap fraction, degree inconsistency is commonly observed between these networks, as illustrated for two networks of similar size, I3D and Tarassov, in Fig. 1a, b. Hubs in one network have small degrees in the other network and vice versa. Although a significant overlap relative to the null model indicates the presence of signal in the network of interest, the small observed overlap fraction resulting from the degree inconsistency makes it appear that most of these networks could be fundamentally different from the reference network. The degree inconsistency becomes even more pronounced when we compare networks of different sizes or modalities, such as PPI networks versus GI networks, as shown in Fig. 1c. In such cases, the raw observed overlap is certainly not a satisfying measure for network agreement since different biological modalities are expected to have very different degree sequences, naturally resulting in a low overlap fraction. This could be resolved by a positive benchmark that takes into account the degree sequences of both networks. Combining the negative benchmark with a positive benchmark can help us to establish a quantitative scale where the observed overlap appears in the context of the worst- and the best-case scenarios.

### Constructing a positive statistical benchmark

A straightforward way to normalize the observed overlap between two networks is to divide the overlap by the number of links in the smaller network[25]. This method is overly optimistic as it takes the size of the smaller network as an upper bound, corresponding to the assumption that all links in the smaller network can be contained by the other network. As an alternative, we first introduce a more realistic 'naïve' upper bound described as follows. The maximum possible overlap is determined by summing up the minimum degrees of each node in the two networks. We then divide the sum by two, as each potential link within a network is considered twice from the perspectives of the two nodes it connects. This naïve upper bound typically yields a much lower reference overlap than the size of the smaller network while still being an exact upper bound, as illustrated in Supplementary Fig. S1a. This finding indicates the dominant role of degree inconsistency in assessing network overlap. However, even the naïve upper bound is not always achievable since it only considers degree constraints independently for each node instead of all degree constraints simultaneously. Thus, a generative model is needed to statistically compare the observed overlap with the upper bound.

To achieve this goal, we propose a generative positive benchmark that takes both the degree inconsistency and the network structure into account. The concept of our positive benchmark can be illustrated through the example of mixing salt and pepper. If we take two jars of salt and mix them together, the resulting mixture will be consistent and identical to the original jars. However, if we replace one of the jars with pepper and mix them together, the resulting mixture will be different from the original jars. When we randomly sample the mixture into two new jars, the new jars will be more similar to each other than the original jars, indicating that the original jars were significantly different. Analogously, we take the union of two initial networks, assuming that they are sampled from the same underlying network. Then, we randomly sample two alternative networks according to the observed degree sequences from the union, utilizing a maximum entropy approach (see "Methods"). The overlap between the two alternative networks forms the positive benchmark, corresponding to the best-case scenario. This way, we generate actual random instances of the best-case scenario with specific overlapping links. The proposed positive benchmark preserves the degree sequence on average during the randomization process (see "Methods"), thereby controlling a major factor that impacts the network overlap. If the two input networks are really sampled from the union, this process builds a statistical ensemble close to the original level of overlap. Instead, if we sample from a complete graph composed of all the nodes of each

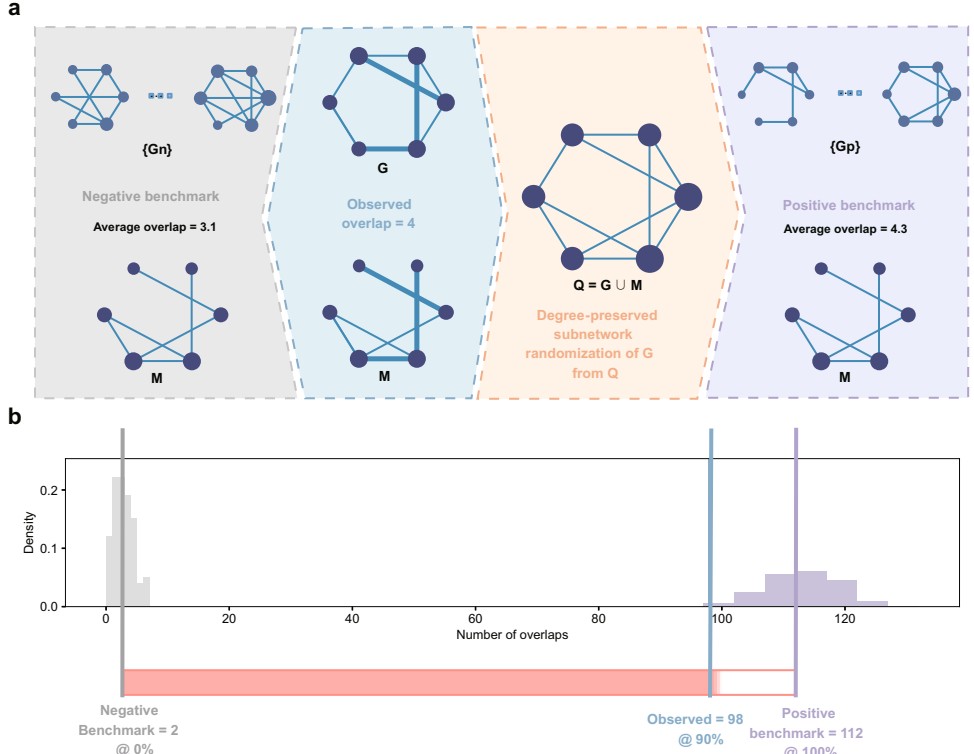

**Fig. 2 | Benchmarks and normalization of network overlap. a** The negative benchmark is obtained by counting the overlap between the network of interest M and the degree-preserved randomized reference networks {Gn}. The positive benchmark is the overlap between M and {Gp}, where {Gp} are obtained by randomly sampling links from the union Q of the reference network G and M according to the degree sequence of G. Thicker links indicate the observed overlap. **b** Example of normalizing the observed overlap (blue) with the negative benchmark (gray) and the positive benchmark (purple) between the reference network I3D (G) and the network of interest, Tarassov (M).

network, i.e., perform degree-preserved network randomization, we arrive at our negative benchmark. In practice, it is sufficient to randomize only the reference network for the positive and negative benchmarks (one-sided statistics, Fig. 2a). In network comparisons without a clear reference network, we randomize one of the networks and calculate the negative and positive benchmarks, respectively (see "Methods"). As expected, the overlap with instances of the positive benchmark is always equal to or lower than the naïve upper bounds with the same degree sequences (Supplementary Fig. S1a) while also being realizable.

Just like testing if the observed overlap is significantly different from the negative benchmark, having access to the positive benchmark allows us to check if the observed overlap is significantly different from the best-case scenario. Comparing 10 yeast PPI networks to I3D shows that the observed overlaps of CCSB-YI1, Ito-core, Uetz-screen, STRING(HsC), and PRS are not significantly different from the positive benchmark, meaning that these networks are in agreement with I3D apart from the observed degree inconsistency (Fig. 1d). We call two networks compatible if the observed overlap is significantly higher than the negative benchmark and not significantly lower than the positive benchmark, meaning that there is little to no room left for biophysical differences or quality issues in the networks apart from the observed degree inconsistency. In contrast, if the observed overlap is significantly lower than the positive benchmark, it means the low observed overlap cannot be explained by the degree inconsistency alone and may indicate the presence of substantially different biological mechanisms or quality issues. For instance, the observed overlap between I3D and STRING(LC) is significantly lower ($p < 0.005$) than the positive benchmark because STRING(LC) consists of low-confidence interactions, most of which are likely false positives.

## Normalized network overlap - Normlap

The positive benchmark, together with the negative benchmark, opens a way to normalize the observed overlap, leading to our definition

$$\text{Normlap score} = \frac{\text{observed overlap} - \text{negative benchmark}}{\text{positive benchmark} - \text{negative benchmark}}, \quad (1)$$

where the *observed overlap* is the number of shared links between the network of interest M and the reference network G. The Normlap score places the observed overlap in the range spanned by the negative and positive benchmarks, respectively. Figure 2b shows an example of normalizing the observed overlap for the Tarassov network with I3D as a reference network.

A major limitation of the observed overlap is its sensitivity to data incompleteness. Thus, we explored if the Normlap score is more robust against incompleteness, potentially providing a less biased view of data quality than the observed overlap. As the most suitable experimental system to test this idea, we considered a systematic proteome-wide human PPI network, HuRI[4], constructed from nine screens, three for each of three different experimental assays. We compared the network of interest, i.e., the network compiled from a number of selected screens from HuRI, with Lit-BM-17[4] as the reference network. As we randomly add more screens to the compiled network of interest, it becomes more complete, leading to a low but steadily increasing overlap fraction from 2 to 5% (Fig. 3a). In contrast, the Normlap scores are less impacted by the incompleteness since the degree inconsistency is already considered, falling within a range of 52–62%, with a slightly decreasing trend as the compiled network becomes more complete. A potential reason for the decreasing Normlap score could be an accumulation of some false positive pairs, slightly reducing the network quality as more screens are combined.

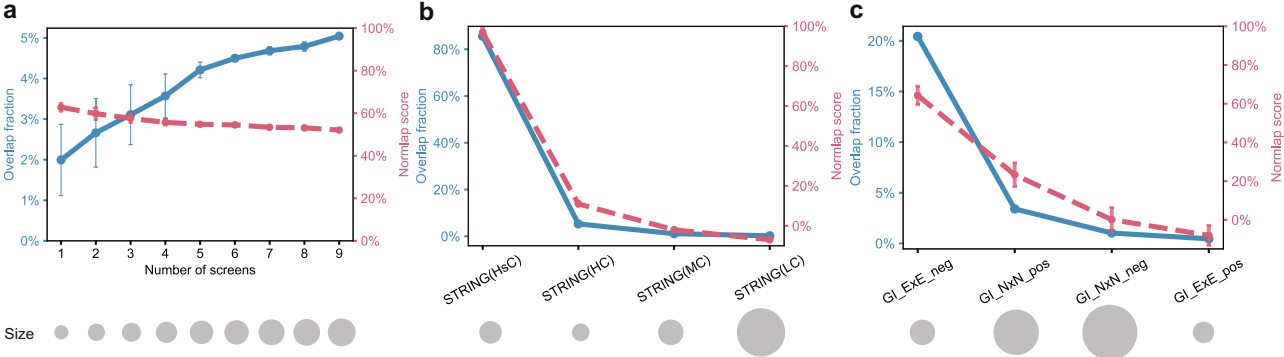

**Fig. 3 | The impact of data incompleteness, quality, and modality on the Normlap score. a** Normlap score and overlap fraction as the number of considered screens increases from the HuRI experiments[4]. The screens are randomly selected. The number of links in each network is indicated by the size of the gray disks. **b** Overlap fraction and Normlap score between I3D and parts of STRING with different confidence score thresholds[24]. **c** Normlap score between the I3D network and GI networks[14]. ExE denotes the interactions between essential genes, and NxN denotes interactions between non-essential genes. Data are presented as mean ± SD.

Since the degree inconsistency is taken into account, the Normlap score is robust not only against data incompleteness but also against changes in the degree distribution, as illustrated in Supplementary Fig. S1b.

We also used yeast datasets to further investigate how the Normlap score changes with network quality by comparing subsets of the STRING dataset of varying confidence levels to I3D as a reference network. We divided PPIs in STRING into four networks according to the confidence score, forming STRING(HsC), STRING(HC), STRING(MC), and STRING(LC) in decreasing order of network quality (see "Methods" for more details). As shown in Fig. 3b, both the overlap fraction and the Normlap score decrease as the network quality becomes lower. However, since a lower overlap fraction could either come from decreased network completeness or network quality, no conclusion about network quality can be drawn from the overlap fraction alone. In contrast, the decreasing Normlap scores indicate a decreasing network quality regardless of the degree inconsistency caused by network incompleteness and other contributing factors.

Next, we used the Normlap score to investigate how well PPI networks agree with each other in yeast. Figure 1d shows the Normlap scores between 10 PPI datasets and the reference network I3D along with the corresponding negative and positive benchmarks and the observed overlaps. All yeast PPI networks scored higher than 89%, except STRING(LC) and RRS. Although the overlap between systematic networks with I3D can be low, the high Normlap scores indicate a high level of agreement between PPI networks. Using the co-complex network as an alternative reference network gives consistent results, as shown in Supplementary Fig. S2.

Comparison of PPI networks to functional networks plays a vital role in understanding the biological rules of a given organism[26]. However, the amplified degree inconsistency (Fig. 1c) between PPI networks and functional networks gets mixed into other factors and leads to a slight overlap at best, making it hard to decipher if the observed difference is due to additional biases or biophysical differences. As a starting point, in yeast, there is a unique genome-wide systematic functional network of genetic interactions[13,14] proven to be highly successful in mapping the wiring diagram of cellular function. Genetic interactions occur between pairs of genes whose simultaneous mutations enhance (positive interactions) or suppress (negative interactions) the fitness compared to the expectation based on the fitness of single mutants. Previous studies[13,14] have shown that negative interactions among essential genes (genes critical for an organism's survival) significantly overlap with PPIs and the co-complex network. This significant overlap is neither observed for non-essential genes nor for positive interactions between essential genes. In addition, positive interactions between non-essential genes overlap with PPIs, albeit to a lesser extent[14].

Here, as an example in yeast, we have utilized the Normlap score to compare the GI networks[14] with four PPI networks and the co-complex network[27,28] (Fig. 3c, Supplementary S2a–d). Although the overlap fraction of negative interactions between essential genes (GI_ExE_neg) with the other five networks varies from 7 to 18%, the Normlap scores are consistently around 60%. This consistency indicates that a large fraction of the GI_ExE_neg network shares compatible biology with PPI and co-complex networks, in line with previous observations[13,14]. For non-essential genes, the overlap fraction of positive interactions (GI_NxN_pos) ranges from 3 to 7%, while the Normlap scores are consistently around 20% except for Lit-BM-20 (12 ± 3%). The consistent Normlap scores indicate that a lesser fraction of the GI_NxN_pos network shares compatible biophysical mechanisms with the PPI and co-complex networks compared to the GI_ExE_neg network. Overall, in the example of yeast and human networks, we showed that the Normlap score is robust against data incompleteness and reflects the agreement of networks reliably, even between networks of different modalities.

## Agreement network of networks

The positive benchmark enables us to provide a comprehensive overview of the knowledge landscape of biophysical networks, even among networks of varying sizes and functions. Here, we present two agreement networks of existing datasets in yeast and human, respectively. In the agreement network, networks are connected if compatible, according to our earlier definition. Figure 4a, c shows the Normlap scores between various networks for yeast and human, followed by the visualization of the agreement networks in Fig. 4b, d.

Our result of yeast datasets shows that all yeast PPI network pairs gained a Normlap score higher than 69%, showing a tendency of compatibility. Note that just like the naïve overlap fraction, the Normlap score is always 100% for a subgraph of the other network, indicating that the subnetwork is consistent with the larger network. For example, CCSB-YI1, Ito-core, Uetz-screen, Tarassov, Sys-NB-06, PRS, and Lit-BM-20 are all subnetworks (or nearly subnetworks) of BioGRID, thus leading to 100% Normlap scores (Fig. 4a). As shown in Fig. 4a, five binary PPI networks, namely YeRI, CCSB-YI1, Ito-core, Uetz-screen, and Tarassov, are compatible with each other. In contrast, these binary PPI networks are not compatible with the non-binary PPI network Sys-NB-06. Since protein pairs in Sys-NB-06 are not necessarily in direct contact, the Sys-NB-06 network has a lower Normlap score (69 ± 4%) with the co-localization network compared to that between PPI networks that require direct or near-direct[11] contact and the co-localization network (86–100%). Besides, most PPI networks and the

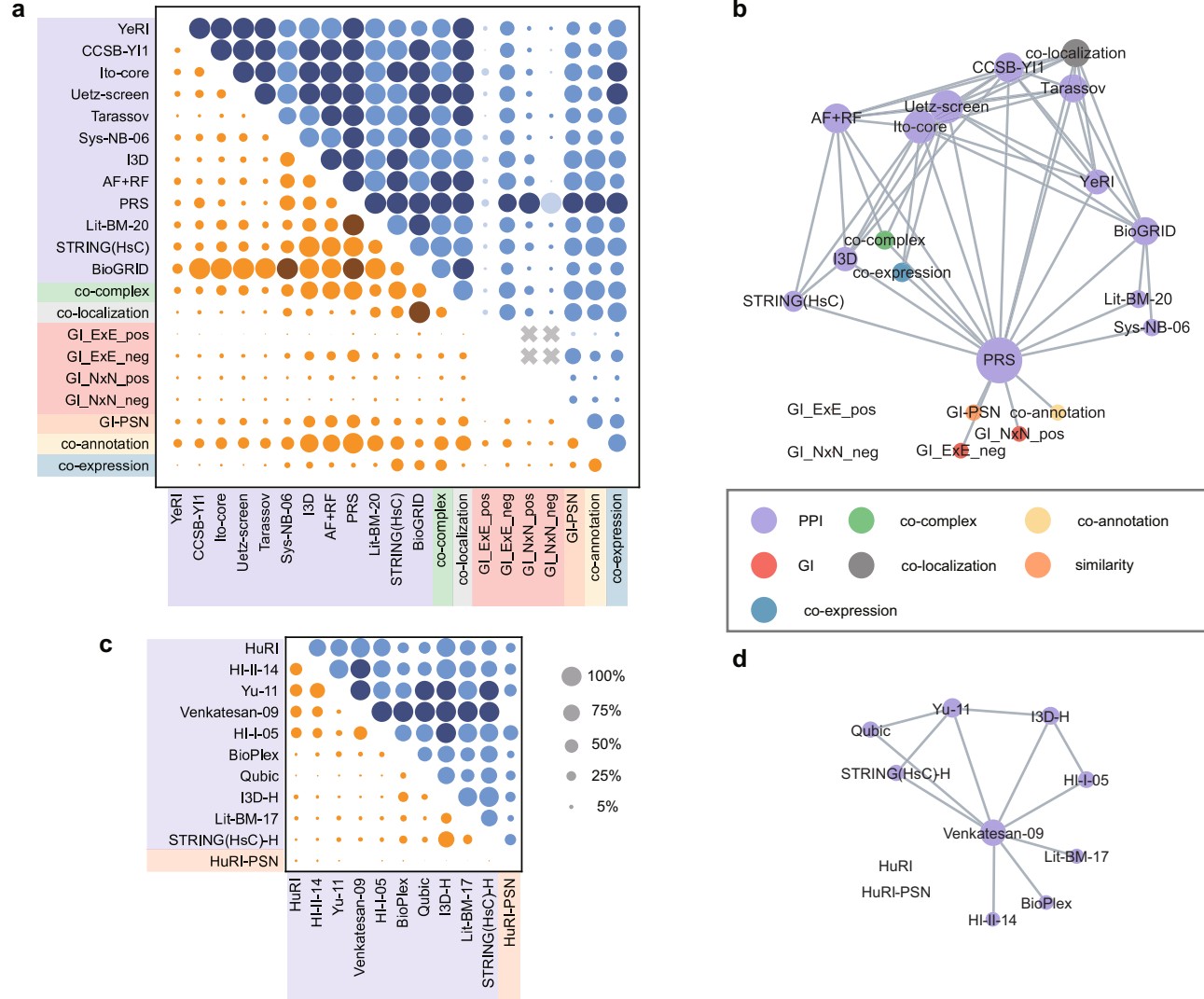

**Fig. 4 | Agreement network of networks.** Normlap score (upper triangle, blue) and observed overlap fraction (lower triangle, brown) between **a** yeast and **c** human networks. Pairs with no significant signal are shown in the lightest color. Compatible network pairs are shown in the darkest color. Pairs that have no common nodes are marked by a gray X in the upper triangle, indicating no conclusions are drawn for these pairs. Agreement network of **b** yeast and **d** human networks. Nodes are connected if compatible. The node size is proportional to the node degree.

co-complex network are not compatible, reflecting the fact that PPIs not only happen within complexes but also between and outside of complexes. YeRI has a slightly lower Normlap score (76 ± 4%) compared to other systematic PPI networks (>89%) when compared to the co-complex network. The lower Normlap score likely originates from the fact that there are more PPIs in YeRI that occur between or outside of complexes, as pointed out in ref. 5. Interestingly, we found that structure-based PPI networks I3D and AF+RF have higher Normlap scores (>84%) than other systematic PPI networks (27–53%) when compared to the co-annotation network. The high Normlap score may be attributed to the fact that proteins with compatible binding interfaces may participate in the same biological process, leading to a higher likelihood of functional associations compared to other PPIs when node degrees are taken into account as a confounding factor.

For human networks, all four Y2H binary networks show a high level of compatibility with the lowest Normlap score at 73 ± 3% between HuRI and Yu-11. As expected, the Normlap score can be much lower when comparing human networks of different experimental methods. For example, the lowest Normlap score of systematic networks is between HuRI and BioPlex (31 ± 1%), indicating a significant difference. The difference may be explained by the fact that BioPlex captures not only binary but also co-complex relationships. Another potential reason for the discrepancy is the observation that Bioplex has significant biases compared to HuRI, which has a more homogeneous data coverage[4,17]. Note that Venkatesan-09 shows compatibility with most human networks for at least two reasons. First, Venkatesan-09 only includes high-confidence PPIs, supported by at least two Y2H reporter assays[15,29]. Second, it is also a small dataset, including only 195 proteins, reducing the statistical power of finding significant differences.

To better understand the potential origins of the observed differences between networks of the same modality, we performed an additional comparison of a separate dataset[30], covering human PPI networks in different cell types, see Supplementary Fig. S4. Compared to the small overlap fractions, the Normlap scores present a more uniform picture. Despite widespread intermediate agreement among most of the networks, some networks displayed notable biophysical differences, such as those obtained in MCF-10A and HEK cells (44 ± 1%). This suggests that the discrepancy between biophysical networks may indeed be partially attributed to varying cellular contexts.

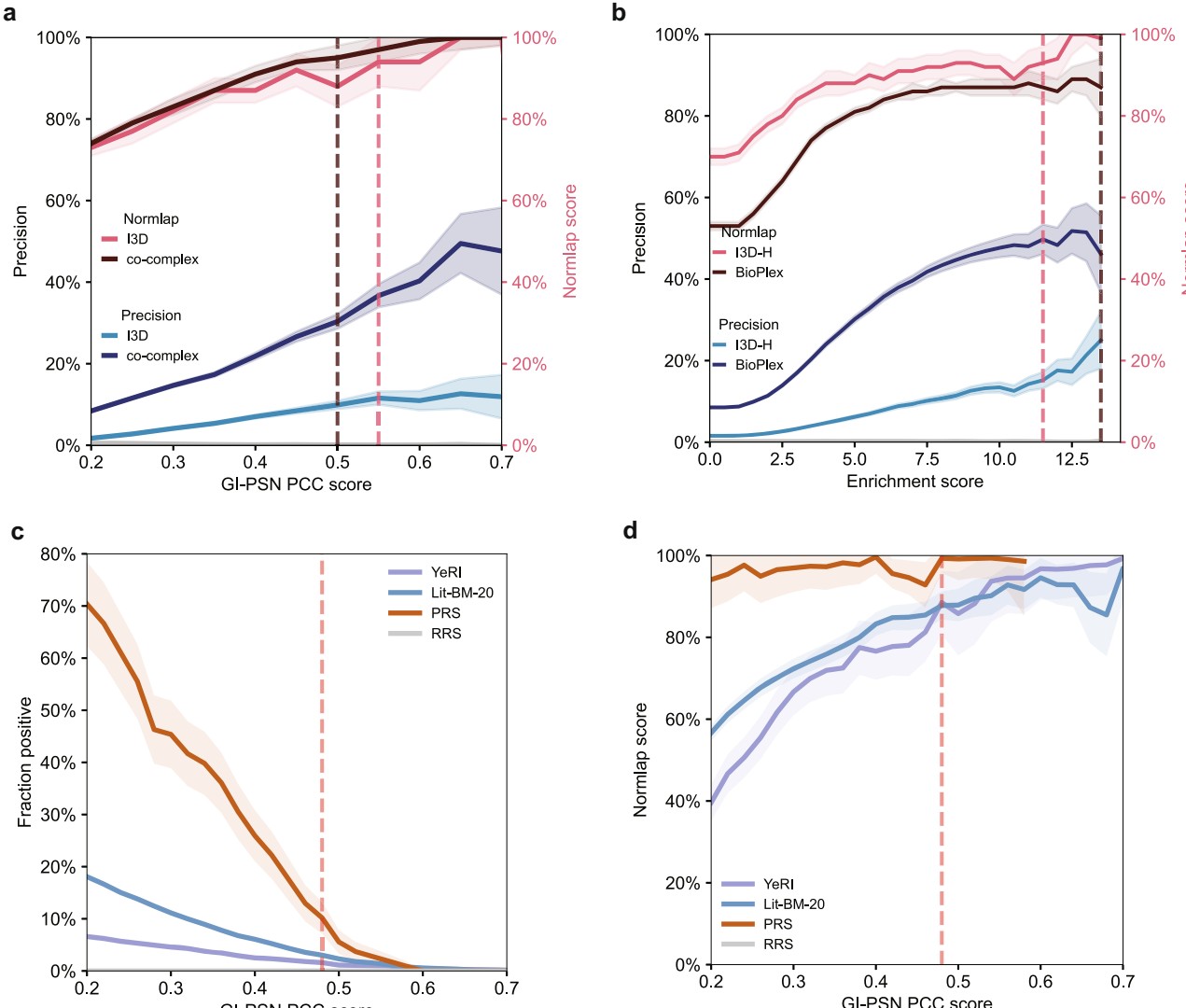

**Fig. 5 | Network evaluation with Normlap.** Validation of **a** the yeast GI-PSN[14] with I3D and co-complex[27,28] networks and **b** the human Qubic[31] with the I3D-H[23] and BioPlex networks[45]. The precision is given by the true positives divided by the total positive predictions. Gray lines indicate the random reference. **c, d** Results of computational validation of YeRI[5] in GI-PSN. Lit-BM-20[5], PRS, and RRS[5] are shown for reference. Red dashed lines indicate the suggested thresholds; see text for details. For Precision and Fraction positive, the shade indicates the error bar determined from shot noise. The Normlap scores are presented as mean ± SD, with the mean shown by the solid lines and SD shown by the shading.

Overall, both agreement networks for yeast and human are well connected, highlighting a high level of compatibility between networks of the same organism. Indeed, we can connect most of the studied network datasets to others, either directly or indirectly, through pairs of compatible networks.

### Thresholding and validating experimental assays

Suppose we have a candidate network where the links are ranked by an intrinsic candidate score. The positive benchmark enables us to threshold such scored networks, making them compatible with a reference network. Indeed, we can compare the candidate network constituted by the ranked links above a selected threshold to a reference network and select the threshold where the filtered network becomes compatible with the reference network. As an illustration, Fig. 5a, b shows the yeast GI-PSN network ranked by Pearson correlation coefficient (PCC) scores[4] and the human Qubic network ranked by enrichment scores[31], respectively. Precision is calculated as $TP/D$, where $TP$ is the overlap, and $D$ is the total number of links with candidate scores above the threshold. The yeast GI-PSN network is compared to two reference networks, I3D and co-complex. We found that the Normlap scores remain consistent for top-ranked links, while the precision varies substantially depending on the selected reference networks. The selected GI-PSN network with PCC = 0.50 (corresponding to 962 links) is compatible with the co-complex network, while I3D gives a threshold at 0.55 (corresponding to 458 links). Turning now to the human Qubic network, which is compared to I3D-H and BioPlex as reference networks, the selected Qubic network is compatible with I3D-H at enrichment score = 11.5 (382 links) and compatible with Bio-Plex at enrichment score = 13.5 (52 links). Therefore, we can systematically threshold candidate networks so that they become compatible with one or multiple reference datasets.

In addition to thresholding scored candidate networks, the positive benchmark allows a computational alternative for validating experimental assays. Experimentally, the quality of new datasets is validated by retesting random links in complementary assays[3–5]. For example, the Y2H v4 screens of YeRI are retested in two complementary binary PPI assays, MAPPIT[20] and GPCA[21], alongside Lit-BM, PRS, and RRS, as shown in Extended Figure 2b in ref. 5. In these

examples, the new dataset is considered high-quality if it has a close positive fraction compared to PRS or Lit-BM at the threshold where the recovery rate of RRS was zero. While retesting with MAPPIT and GPCA assays provides a way to validate the experimental datasets, it comes with sizeable costs to retest batches of the experimental data in different labs. Besides, the positive fraction is expected to depend sensitively on the degree inconsistency with the complementary assays and whether the tested links are within complexes or outside of complexes (Fig. 2b, c in ref. 5). Furthermore, due to the high cost there is currently no MAPPIT or GPCA network to systematically assess and address the degree inconsistency.

As a computational alternative, we propose validating new experimental datasets with an existing scored network. First, the links in the new datasets are ranked by the scores in the existing network. Then, we generate the positive benchmark from the union of the new dataset and pairs above a selected threshold in the scored network. Finally, we choose the threshold where the new dataset is compatible with the scored network at the selected threshold. As a proof of concept, we validated YeRI with the GI-PSN network (Fig. 5c, d), along with Lit-BM-20, PRS, and RRS, for reference. Links in the four networks are ranked by the PCC scores in the GI-PSN network, and at PCC = 0.48, YeRI is compatible with the GI-PSN network. Just like in the experimental validation with MAPPIT or GPCA assays, we have comparable normalized positive fractions of the validated network YeRI ($89 \pm 8\%$) with Lit-BM-20 ($88 \pm 3\%$) and PRS ($99 \pm 4\%$) while RRS is 0% at the selected threshold, serving as validation of the YeRI methodology. Note that we could choose a lower threshold as long as the RRS is sufficiently low (-0%).

## Discussion

Complex biological networks require multiple complementary approaches to be fully mapped and characterized, inevitably leading to datasets of low overlap. While a significant overlap compared to a negative benchmark shows signal in the dataset, the observed overlap is not a reliable indicator of network agreement due to multiple factors contributing to the degree inconsistency. To better assess the network agreement, we proposed a generative model for a positive statistical benchmark that takes into account the degree inconsistency as a confounding factor. The positive benchmark, together with the negative benchmark, provides the necessary context for the observed overlap, leading to our normalized network overlap score—Normlap.

We demonstrated that the Normlap score is robust to data incompleteness and capable of reflecting network agreement, with the example of human and yeast networks. Furthermore, the positive benchmark and Normlap score enable a standardized comparison of different molecular and functional networks. As a case study, we evaluated the agreement between various yeast and human network datasets. In contrast to the low observed overlap, we observed widespread compatibility among PPI networks of the same organism, as most datasets form a cluster of compatible networks. Although we mostly used I3D and I3D-H as reference networks, other networks may serve as reference networks. Just like there is no single superior assay to map biological networks[12], there might not be a universal reference network either. Yet, our work can be an important step toward creating the next generation of PRS datasets. Finally, as a computational alternative to experimentally retesting existing networks, we demonstrated how to threshold and validate experimental assays using the positive benchmark and Normlap score.

A limitation of the proposed positive benchmark is that it is overestimated for the following reasons. First, we take the union of the two networks as a proxy for the ideal network, while the ideal network can be more complete, leading to a smaller overlap. As a potential solution, the union can be made more complete to gain a better estimate of the true underlying network. Instead of taking the union of two networks, we can add more networks to the union if it is reasonable to

assume these networks originate from the same underlying biology. Second, all the links in the union are assumed to be true positives, while the union likely includes some noise. The impact of possible noise can be illustrated by considering the extreme case where one network is randomized. Even in this case, the positive benchmark will be higher than the negative benchmark (Supplementary Fig. S1b), meaning that treating all links in the union as true positives, in this case, leads to an overestimated positive benchmark. More generally, the Normlap score depends non-linearly on the amount of noise in the datasets. Therefore, additional steps are needed to reliably quantify the noise level of the networks through a process of systematic calibration. Yet, the typically observed high Normlap scores between networks of the same modality, such as in the case of PPI networks in Fig. 1d, leave little to no room for significant noise in the studied datasets, as the low observed overlap is well captured by differences in the reported node degrees.

Our framework has the potential to facilitate new biological discovery in multiple ways. First, it can enhance protein functional annotation inference by improving the search for new assays of biological networks, such as PPI networks, in which connectivity patterns can reveal protein functional annotations[4]. Our framework can also be used as a stand-alone computational tool or as part of existing experimental benchmarking procedures to identify high-quality assays, even if designed to be complementary with each other, leading to a more complete interaction map and thus enhancing protein functional annotation inference. Second, our method of assessing network agreement allows us to identify biological differences between different experimental conditions or assays. Currently, most networks exhibit low overlap due to degree inconsistencies. Our comparison can identify the presence of additional biophysical and/or quality differences between networks beyond degree differences. For example, our results show that Y2H assays such as HuRI exhibit a different picture of interactions between mammalian proteins from BioPlex and Qubic. Once such differences have been identified, they can be further explored through various follow-up experiments. Finally, our framework can improve interactome size estimates. Existing estimators rely on the network overlap[32] without taking into account the degree inconsistency as a confounding factor. However, using networks that are incompatible with our definition can lead to an overestimation of the interactome size. The reason is that incompatibility indicates biophysical differences or quality issues that are not addressed by incompleteness alone. Utilizing compatible networks can therefore improve the accuracy of the estimates. More broadly, our framework has far-reaching potential applications across a spectrum of domains that involve the assessment of networks, such as various brain[33] or social networks[34]. The methodology of generating the positive benchmark can be further extended to more complicated networks such as directed, signed or weighted networks[35], multi-layer networks[36], and dynamical networks[37].

## Methods
### Network construction
Yeast gene names have been mapped to the ORFs (open reading frames) with the SGD database[38]. Genes that could not be mapped to nuclear ORFs have been excluded from our analysis. For consistency among datasets, we have removed the hyphens in the ORFs so that ORF like "YDR374W-A" became "YDR374WA". Human datasets were mapped via gene or protein identifiers to the Ensembl gene ID space with the hORFeome Database 9.1[39]. Genes that cannot be mapped to Ensemble gene ID are removed from this analysis. All self-interactions in yeast and human datasets are excluded from this analysis. Summaries of the number of nodes and links for all yeast and human datasets used are shown in Supplementary Tables S1 and S2.

## Yeast PPI networks

**Systematic Y2H PPI networks—YeRI, CCSB-YI1, Ito-core, Uetz-screen.** YeRI is the latest systematic map constructed using a novel Y2H assay version after testing ~99% of the yeast proteome[5] three times with assay Y2H v4. CCSB-YI1[3] is an earlier proteome-scale dataset of Y2H PPIs validated using the two complementary assays, MAPPIT[20] and GPCA[21]. Ito-core[6] is a subset of PPIs found three times or more in Ito et al.[6], excluding unreliable pairs of proteins found only once or twice. Uetz-screen[5] is a subset of PPIs from Uetz et al.[7] that was obtained from a proteome-scale systematic Y2H screen, excluding a smaller-scale, relatively biased, targeted experiment with a smaller number of well-studied bait proteins.

**Physically proximal PPI network—Tarassov.** The Tarassov network is a proteome-scale dataset generated using a dihydrofolate reductase protein-fragment complementation assay (DHFR PCA) by Tarassov et al.[11]. It contains physically proximal but not necessarily directly contacting protein pairs.

**Systematic AP-MS PPI network—Sys-NB-06.** The Sys-NB-06 dataset is obtained from ref. 5, which contains PPIs from three AP-MS experiments, namely Gavin 2002[8], Gavin 2006[9], and Krogan 2006[10].

**Inferred PPI network with experimental structures—I3D.** The I3D dataset used in this analysis is a subset of Interactome3D[23], released in May 2020. The subset is restricted to experimental structures, with interactions from homology models excluded. Proteins with compatible binding interfaces are identified as PPIs in I3D.

**Predicted PPI network from structures—AF+RF.** AlphaFold (AF)+RoseTTAFold (RF) is a deep-learning-based predicted PPI network, downloaded from ref. 5. The subset used in this analysis is restricted to links with PPI score ≥0.9.

**Literature curated datasets—Lit-BM-20.** The Lit-BM-20 dataset is from ref. 5. Links with two or more pieces of evidence, including at least one binary evidence, were selected as Lit-BM-20 links.

**Positive reference set (PRS) and Random reference set (RRS).** The PRS and RRS dataset is from Lambourne et al.[5].

**BioGRID.** The BioGRID yeast PPI dataset is constructed from the 4.4.210 release of the BioGRID database[40]. The Organism ID and Experimental System Type are set to be "559292" and "physical".

**STRING.** The STRING yeast PPI network is constructed from the v11.5 release STRING database[24]. The physical subnetwork is segmented into four networks with different confidence levels, namely highest confidence network STRING(HsC) (quality score ≥ 900), high confidence network STRING(HC) (700 ≤ quality score < 900), medium confidence network STRING(MC) (400 ≤ quality score < 700) and low confidence network STRING(LC) (150 ≤ quality score < 400).

## Yeast functional networks

**Yeast genetic interaction (GI) networks.** Genetic interaction networks are constructed from the SGA data by Costanzo et al.[14]. The source data are used at the intermediate threshold ($p < 0.05$ and genetic interaction score $|\epsilon| > 0.08$). For interactions with multiple $p$-value and $\epsilon$, the $\epsilon$ corresponding to the smallest $P$ is used for classification. The same process is applied to both essential (ExE) and non-essential (NxN) gene interaction source datasets to construct the GI_ExE_pos, GI_ExE_neg, GI_NxN_pos, and GI_NxN_neg networks.

**Yeast genetic similarity network.** The genetic similarity networks are constructed from the genetic interaction profile similarity matrices by Costanzo et al.[14]. The dataset is filtered with PCC > 0.2 to construct the genetic similarity network (GI-PSN).

**Yeast co-complex network.** The yeast co-complex network was generated from the list of protein complexes from Baryshnikova 2010[27] and Benschop 2010[28]. Genes in the same complex are connected. Note that 495 genes are classified into multiple complexes, leading to bridges between complexes.

**Yeast co-localization network.** The yeast co-localization network is constructed from the BioGRID[40] database (4.4.210 release). The Organism ID and Experimental System are set to "559292" and "co-localization" to filter for yeast co-localization links.

**Yeast co-annotation network.** Genes are considered co-annotated if they share annotation from the non-redundant set of specific GO terms[41].

**Yeast co-expression network.** The co-expression data are from https://coxpresdb.jp[42]. The union dataset Sce-u.v21 was used. Union co-expression is calculated by the average of the logit-transformed MR values, which is the measure of co-expression strength in COXPRESdb, of RNAseq and microarray co-expression; for gene pairs with only RNAseq co-expression, RNAseq co-expression values were converted to union values by linear regression. Only pairs with scores ranked in the top 5000 were used to generate the co-expression network.

## Human networks

**Systematic Y2H PPI networks—HuRI, HI-II-14, Yu-11, Venkatesan-09, HI-I-05.** HuRI[4] is a human 'all-by-all' reference interactome map of human binary protein interactions from nine Y2H screenings of ~17,500 × 17,500 proteins. HI-II-14[43] included binary PPIs from the Y2H screen for interactions within a "Space II" matrix of ~13,000 × 13,000 ORFs contained in Human ORFeome v5.1. Yu-11[44] tested ~6000 × 6000 ORF search space of human ORFs in the ORFeome 3.1 with Y2H screening. Venkatesan-09[15] contains high-quality Y2H interactions from four Y2H screens that were performed on a set of ~1800 × 1800 protein pairs that were initially designed to estimate the coverage and size of the human interactome. HI-I-05[29] is the first map of the human binary interactome obtained by Y2H screening for direct, binary interactions within a "Space I" matrix of ~8000 × 8000 ORFs contained in Human ORFeome v1.1.

**AP-MS PPI network—BioPlex.** BioPlex[45] is a proteome-scale, cell-line-specific interaction network. It results from affinity purification of 10,128 human proteins−half the proteome−in 293T cells. The BioPlex v3.0 data were downloaded from https://bioplex.hms.harvard.edu on February 14th, 2022.

**Inferred PPI network with experimental structures—I3D-H.** The I3D dataset used in this analysis is a subset of Interactome3D[23] released in May 2020. PPIs are restricted to interactions between human proteins.

**Qubic.** Qubic is a proteome-wide human interactome with the quantitative BAC-GFP interactomics (QUBIC) method[31]. Individual interactions are characterized in three quantitative dimensions that address the statistical significance, interaction stoichiometry, and cellular abundances of interactions.

**Literature curated dataset—Lit-BM-17.** The Lit-BM-17 dataset is from Luck et al.[4]. Each PPI with multiple pieces of evidence, with at least one corresponding to a binary method, is annotated as Lit-BM-17.

**STRING(HsC)-H.** The STRING human PPI network is constructed from the v11.5 release STRING database[24]. The network is restricted to the

physical subnetwork among human proteins and to the PPIs with the highest confidence score (≥900).

**Human similarity network—HuRI-PSN.** Jaccard similarity (number of shared interaction partners divided by the total number of interaction partners) was calculated for every pair of proteins of degree ≥2 in HuRI[4].

**Cell-line-specific human PPI networks.** In this dataset, literature-curated PPIs are annotated with cellular context information, which is derived through literature data mining[30]. Here, we focus on the 20 cell lines with the most PPIs.

### Construction of positive and negative benchmarks based on maximum entropy framework

The positive and negative benchmarks are constructed based on the maximum entropy framework, which enables us to efficiently incorporate both hard and soft constraints. The entropy is maximized subject to these constraints, producing the most random distribution that is still in compliance with the considered constraints.

We consider two graphs $G(V_1, E_1)$ and $M(V_2, E_2)$. For the positive benchmark, G and M are combined to form $Q(V_1 \cup V_2, E_1 \cup E_2)$, which includes all nodes and links from G and M. The graph ensemble {Gp} contains subgraphs of Q and is constructed by the maximum entropy approach to preserve the expected value of the node degrees in G[22]. For example, to construct {Gp}, we fix the mean subgraph node degree at its original value $<k_i>_{Gp} = k_i(G)$. The resulting probability of having a link between nodes $i$ and $j$ is given by $p_{ij} = 1/(1 + \alpha_i \alpha_j)$, with the expected subgraph degree of a node $\langle k_i \rangle = \sum_{j,(i,j) \in Q} 1/(1 + \alpha_i \alpha_j)$. The optimized $\alpha_i$ can be found iteratively with the update rule $\alpha'_i = \frac{1}{k_i} \sum_{j,(i,j) \in Q} 1/(\alpha_j + 1/\alpha_i)$. For the negative benchmark, G is randomized similarly, except that Q is replaced by the complete graph of nodes in G. We start with the initial condition $\alpha_i^{(0)} \equiv 1$ and perform a large number of iterations to update $\alpha_i$. For the negative benchmark, we stop the iterations when the maximum relative change of $\alpha_i$ is less than $10^{-6}$. For the positive benchmark, $\alpha_i$ converges slower compared to the negative benchmark due to the high number of constraints. In this case, we stop iterating $\alpha_i$ if the change in the mean overlap of the positive benchmark is less than 1 during the last 1000 iterations. $\alpha_i$ are then used to calculate the connection probability $p_{ij}$ for all links in Q. We can calculate the average and variance of the (positive or negative) benchmark as $\Sigma_{(i,j) \in M} p_{ij}$ and $\Sigma_{(i,j) \in M} p_{ij}(1 - p_{ij})$, without the need to explicitly generate random samples from the ensemble. This enables an efficient evaluation of statistical significance[22].

In the case where there is no clear reference network, we randomized either of the networks. The absolute z-score is calculated for the benchmarks and the observed overlap as

$$z = \frac{|\text{mean benchmark} - \text{observed overlap}|}{\text{SD of benchmark}}. \quad (2)$$

The negative (positive) benchmark that has the lower absolute z-score is used as the final negative (positive) benchmark to calculate the Normlap score.

We transformed the z-score to a one-sided p-value using the cumulative density function of the standard normal to check if the observed overlap is statistically higher than the negative benchmark or lower than the positive benchmark. A $p < 0.05$ value indicates that the observed overlap is significantly lower (higher) than the positive (negative) benchmark at a confidence level of 95%. All algorithms are implemented in Python 3.10.6 with numpy, pandas, and scipy packages.

### Visualization of agreement networks

We applied the relative entropy optimization (EntOpt 2.1)[46] layout to the agreement network in Cytoscape 3.7.1[47]. In the agreement

networks, compatible networks are connected, indicating that there is no sign of biophysical differences or quality issues between the two networks. For GI networks between essential genes and non-essential genes, both observed overlaps and positive benchmarks are 0 since they have no common nodes. In such cases, no conclusions are made, and the corresponding links are not shown in the agreement network.

### Reporting summary

Further information on research design is available in the Nature Portfolio Reporting Summary linked to this article.

## Data availability

The analyzed data generated in this study are provided in the Source Data file. The data used in this study are available at https://github.com/hbj153/normlap_paper or under Zenodo https://doi.org/10.5281/zenodo.7872611. Large-scale data can be downloaded from the Bio-GRID (https://thebiogrid.org) and STRING (https://string-db.org) databases. Source data are provided with this paper.

## Code availability

A repository containing a Python package for calculating the positive (negative) benchmark and generating instances is available at https://github.com/hbj153/normlap or under Zenodo https://doi.org/10.5281/zenodo.7872625.

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

## Acknowledgements

We thank L. Lambourne for helpful correspondence and R. Xie, H. S. Ansell, and R. Chepuri for useful discussions.

## Author contributions

Theoretical study and data analyses: I.A.K. and B.H. Writing: I.A.K. and B.H. Project design: I.A.K. and B.H.

## Competing interests

The authors declare no competing interests.
