## [Peer Review File · Nature Communications]

REVIEWER COMMENTS

Reviewer #1 (Remarks to the Author):

Report on Hao and Kovacs „A positive statistical benchmark to assess network agreement”

Hao and Kovacs point out that comparison of protein networks is often not intuitive, in the sense that the overlap between networks, though highly significant, is often low in absolute numbers. Therefore overlap assessment is often not a good quality measure in network comparison/benchmarking.

They propose to use a best-case network scenario in and a full wired negative data set to scale what is called a “Normlap” value. In brief the idea is to construct a union of the network as a best-case benchmark network and use the fully wired complement as negative benchmark. Estimating the overlap through rewiring of the two network gives a statistical overlap of negative and positive data that is then used to scale the real overlap. To my knowledge the idea is novel and can be very useful as a benchmark measure in the network biology field.

The idea is exemplified with yeast and human protein interaction and genetic interaction data. It seems as if the Normlap score gives a very robust estimate, which is nicely shown in Figure 3a using human Y2H repeat screening data (overlap at about 3-5%). However, it does not deal well with large overlaps, and the use of data sets is selective.

Points for consideration

*) There is no demonstration of outcome, is there anything new to learn about the networks trough using this score?

*) Conceptionally, degree inconsistency as the source of limitation of overlap is strongly put forward. I do not see any data that actually show that degree inconsistency is the problem of overlap analyses. In particular if JJ indices are used (many studies do so) than high degree nodes can impact negatively and positively. In general, it is clear, degree is the most confounding factor in any network analysis. In other words, how dependent is the overlap on the degree distribution. How dependent is the NormLap on the degree distribution? How dependent is the score on removal of high degree nodes?

*) Clearly the paper is written in an entertaining way. However, some of the explanations do not help. E.g. the salt and pepper comparison does not appeal to me.

Also this sentence for example is not useful. “The negative (positive) benchmark that is closer to the observed overlap is used as the final negative (positive) benchmark.” I guess the criteria applies for the negative and the positive network when only one of the networks is randomized.

*) Use of networks and benchmark is very selective. I3D is the only benchmark (and contains the network data that are then compared).

*) Using human Y2H from the same lab, the Vidal lab, may actually address reproducibility than actual network overlap. The HI-I-05 is included in HI-II-14 and this is included in HuRI. To a large extent (maybe unless clearly different experimental strategies were used) this comparison of the data does not make sense. In particular if they underwent pairwise retesting.

*) Figure 4d HURI is not connected, why is that the matrix shows strong connections. Why is Venkatesan a central node in the graph?

Reviewer #2 (Remarks to the Author):

Review 1 of: A positive statistical benchmark to assess network agreement

The authors present a computational framework to assess overlap among networks. While the approach is conceptually interesting and innovative, it is not clear that the calculations are suited for the claims. Also, the discussion of problem and other approaches, although informed, is not entirely up to the state of the art in the field.

This is exemplified by the first sentence of the abstract stating that 'significant overlap' with a reference network was 'current best practice'. As the manuscript is largely focused on protein interactions, this is not correct. Current best practice is to test a random subset from the new dataset in a benchmarked orthogonal validation assay.

Importantly, as the authors point out, there is no inherent measure to dataset precision, and false positives in both networks are handled as 'true positives'. At the end, the authors make a revealing point by suggesting that their normlap score depends on noise (i.e. poor quality) and that reliable assessment of noise (dataset quality) requires other approaches. Thus, even in the manuscript the authors agree that their approach is not suited for quality control as insinuated in the abstract and throughout the text.

How do the authors handle search space issues (see Venkatesan et al, 2009): if we have two perfect representations of biology but for non-overlapping sets of 200 genes, each, there will be no overlap, thus suggesting, to anyone thinking that overlap had any relation to data quality, that this must be low. It is not clear how this scenario was handled in the process of forming a union and subsequently splitting it again.

Ln 29: yes, no assay can capture all interactions but this is a statement about a single biological modality, when switching biological modalities, this is a different statement, which is at the same time trivial and not an 'example' of the first statement.

Ln 30. PCA is an assay that interrogates 2 interaction partners, but is not a 'binary' assay in a sense of detecting predominantly direct interactions. In fact, I believe the QC data in the manuscript suggest the data are closer to co-complex datasets - please double check.

Ln 44: no, the question if overlap suggests data quality has been addressed and it is not unless many aspects of the dataset are taken into account – including completeness and precision. The authors fully omit discussion of the interactome mapping framework (Venkatesan et al. Nat Meth 2009), and prior work on the yeast interactome Yu et al, Science 2008, and QC approaches based on confidence scoring as a state-of-the-art for quality control (Braun et al, Nat Meth 2009).

Ln 64: why would one expect or need a consistent degree sequence – the authors discuss completely different biological modalities.

Ln 88: Please justify why I3D is taken as 'gold standard'. Due to its nature as being supported by structures one would expect this to be similarly biased and incomplete, no? Please rephrase and justify the use as a reference (which I strongly suggest as alternative to 'gold standard') despite such problems.

Ln 102: 'small overlap' between networks of different biological modalities (GI, PPI) could suggest that 'most of these networks are problematic' – this phrase is aggravating. How could any person expect a large overlap between networks in different biological modalities and even make a strong assumption to this extent to then conclude from the lack thereof, that this is reflective of a poor data quality.

I do agree with the authors that assessment of overlap can be useful or interesting, but phrasing an overlap analysis as a way to assess quality of one or the other compared networks even across biological modalities is, my apologies, ignorant.

Ln 129: It is not fully clear what the authors are doing. Forming the union of two networks is understood, but it is unclear how a split of this union into two networks yields any overlap. How do the authors handle double edges, are they kept as two or collapsed into one? Please specify the approach.

Also, conceptually, it is not clear how the approach to mix and split is justified. Partly this results from the confusion as to how the overlap among two partitions of one union network yields any overlap. But also, it is not clear, how mixing of networks from different biological modalities, or of binary and co-complex interactions is justified and meaningful.

Ln 130: please be specific, which complete graph is available?

Ln 181 - 183: yes, understanding different modalities is important, but a degree inconsistency does not hinder further interpretation but reflects difference centrality of a protein in different biological modalities – a TF regulating thousands of genes, is not expected to also be a central hub in the metabolic network. So, different degree distributions do NOT preclude further interpretation.

185: the genetic network is not mapping the wiring diagram – please rephrase

Ln 216 and following: what exactly do you mean to express by 'compatible' in biological terms.

Ln 228: structurally supported interactions could have informed functional annotations, or merely act in the same processes, e.g. SH2 domains, SH3 domains, death domains. Please keep this circularity in mind in the discussion.

Ln 257: No, dataset quality is NOT assessed by testing promising links, but a random sample. This is critical.

Ln 263: the 'degree inconsistency' is a consequence of different biochemical biases of different assays, and this is captured by the benchmarking. Therefore the approach is the only way to assess data quality, and better so than any overlap. Please check the relevant literature and rephrase (Yu, 2008, Venkatesan 2009, Braun 2009).

Reviewer #3 (Remarks to the Author):

The article "a positive statistical benchmark to assess network agreement" written by Hao and Kovacs introduced a new generative positive benchmark approach and associated evaluation metric score (Normlap score) for quantitatively assess the network quality. The paper is well written with enough conceptual description and examples. The idea of mixing network and re-sampling is conceptually simple but neat. I have the following questions regarding this paper, and hope they could be addressed before publication.

1. In the section of "Agreement networks of networks", the little agreement between HuRI and BioPlex is surprising but maybe expected. Many papers claimed that yeast-hybrid approach in general has a very high false positive rate as the hybridizing reaction happens in yeast nucleus that is not the original cellular condition. Could authors add one section to benchmark the two networks using I3D-H, I am very curious to see the evaluation result between HuRI/BioPlex with I3D-H using the new metric.

2. In the section of "Threshold and validate experimental assays", the authors tried to validate yeast GI-PSN to I3D and co-complex networks. To me, this doesn't make sense, GI-PSN is genetic interaction network. Genetics interaction is very complicated itself, where the two genes might function in parallel pathways, in same complex/pathway, in cross-talked pathways...To compare genetics network with physical pathways are like comparing apple to orange.

3. In the section of "Normalized network overlap", the authors again discussed using Normlap score to compare GI network with four PPI network and co-complex network. First, I don't think the comparison is meaningful as genetic network and physical network are apple and orange. Second, they claimed

overlap fraction of GI_NxN_neg is much smaller than GI_ExE_neg. I don't see how the claim draws any meaningful biological conclusion, as the size of GI_NxN_neg is much larger than GI_ExE_neg.

4. One conceptual concern is the biological contexts/cellular contexts are not discussed in this paper. For example, networks mapped from different cell lines/conditions...are of course different. Could the authors make some comments about this and their newly developed approach?

5. The idea introduced in this paper is theoretically sound and interesting with many abstract/conceptual examples. But I don't see a clear way to use it for facilitating new biology discovery, which I think this is the final goal of biological network mapping. Thus, I think the authors should add a section to discuss/demonstrate how to use the new statistical framework to help new biology discovery.

6. There are a lot of p-values listed in this paper. The names of statistical tests are usually missing and I don't have a clear idea how these p-values are computed. The authors should clarify them in the revised version.

7. In the abstract, the authors mentioned this is a "maximum entropy framework", which is confusing. I don't see the connection between the approach introduced in this paper and information theory. The authors should clarify this in the revised version.

8. The colors used in this paper are bad. It is hard for me distinguish lines in plots (such as Figure 3, 5, S2). The authors should print the paper and see how difficult it is for readers to read these plots. So, colors should be changed and lines should be made thicker in all plots.

RESPONSE TO THE REVIEWER COMMENTS

We appreciate the time and effort of the Reviewers in providing constructive feedback on our manuscript. These thoughtful suggestions have prompted us to significantly improve the presentation of our results. Appended is our point-to-point response to the comments raised by the Reviewers, with our responses in blue color. We have also submitted a revised manuscript with changes shown in blue.

Reviewer #1:

Hao and Kovacs point out that comparison of protein networks is often not intuitive, in the sense that the overlap between networks, though highly significant, is often low in absolute numbers. Therefore overlap assessment is often not a good quality measure in network comparison/benchmarking. They propose to use a best-case network scenario in and a full wired negative data set to scale what is called a “Normlap” value. In brief the idea is to construct a union of the network as a best-case benchmark network and use the fully wired complement as negative benchmark. Estimating the overlap through rewiring of the two network gives a statistical overlap of negative and positive data that is then used to scale the real overlap. To my knowledge the idea is novel and can be very useful as a benchmark measure in the network biology field. The idea is exemplified with yeast and human protein interaction and genetic interaction data. It seems as if the Normlap score gives a very robust estimate, which is nicely shown in Figure 3a using human Y2H repeat screening data (overlap at about 3-5%). However, it does not deal well with large overlaps, and the use of data sets is selective.

Points for consideration

1.1) There is no demonstration of outcome, is there anything new to learn about the networks through using this score?

Probably our most striking discovery using Normlap is that the observed, seemingly low, overlap between protein-protein interaction networks is typically as good as it can get, indicating a high level of agreement. Thus, surprisingly, apart from obvious differences at the level of node degrees, there is no major difference between most of these “complementary” networks, meaning there isn’t much room for errors or quality issues. As modeling noise and biases for each of these experimental techniques is an ongoing challenge, our finding is an important indication of a very positive message that these various complementary methods are capturing high-quality and consistent biophysics. As for demonstration of outcomes, we included two case studies in Fig.4 on the yeast and human networks, showing how our methodology can contribute to an improved way to assess new experimental assays as well as the biological content of functional datasets. As for the question of what we can learn about each network by doing our improved comparisons, see our answer to point 3.2.

1.2) Conceptionally, degree inconsistency as the source of limitation of overlap is strongly put forward. I do not see any data that actually show that degree inconsistency is the problem of overlap analyses. In particular if JJ indices are used (many studies do so) than high degree nodes can impact negatively and positively. In general, it is clear, degree is the most confounding factor in any network analysis. In other

words, how dependent is the overlap on the degree distribution. How dependent is the NormLap on the degree distribution? How dependent is the score on removal of high degree nodes?

The very fact that the Normlap score is often close to 100% is a clear indication that the major limitation of overlap is degree inconsistency between protein-protein interaction networks. We also show explicitly that the degree inconsistency alone seriously limits the potential overlap, even before introducing Normlap, through the naive upper bound in Figure S1b.

Specifically, to show that the proposed Normlap score is independent of degree distribution, we have included a new figure (Fig. S1b) demonstrating how the overlap fraction and Normlap scores change when high degree nodes are removed in Tarassov, in comparison to I3D as a reference network. The corresponding discussion is added to the main text in lines 187-189. The main point is that the Normlap score is robust against the removal of high degree nodes since the degree inconsistency is already taken into account as a confounding factor.

1.3) Clearly the paper is written in an entertaining way. However, some of the explanations do not help. E.g. the salt and pepper comparison does not appeal to me. Also this sentence for example is not useful. "The negative (positive) benchmark that is closer to the observed overlap is used as the final negative (positive) benchmark." I guess the criteria applies for the negative and the positive network when only one of the networks is randomized.

We appreciate the Reviewer's suggestion on improving the readability of the manuscript, and we have rephrased some explanations accordingly. As for the quoted sentence, the Reviewer was absolutely right and we moved this note to the methods section.

1.4) Use of networks and benchmark is very selective. I3D is the only benchmark (and contains the network data that are then compared).

We have included additional text to explain why we choose I3D as one of the reference networks in this study (see lines 94-98). Ultimately, the right choice of a reference must depend on the biological context and the questions addressed. We are not claiming that I3D is the best such reference, it is just one example to show that our methodology improves on how we compare to reference networks. Note that I3D is not the only reference network we used in this paper:

1. GI networks are compared to five different references (Fig. 3d and Fig. S2a-d).
2. In the 'Threshold and validate experimental assays' section, we use other networks as references, for example the co-complex data in yeast data and BioPlex in humans.

To further illustrate that our main conclusions are not sensitive to the choice of a benchmark network, we added a supporting figure that compares yeast PPI networks with the co-complex network as a reference in Fig.S2 and lines 203-204. With the ongoing development of new methods and datasets, we anticipate that the field will converge on more suitable reference networks. We believe that our methodology will play a role in identifying such references, as going beyond degree constraints is an important step.

1.5) Using human Y2H from the same lab, the Vidal lab, may actually address reproducibility than actual network overlap. The HI-I-05 is included in HI-II-14 and this is included in HuRI. To a large extent (maybe

unless clearly different experimental strategies were used) this comparison of the data does not make sense. In particular if they underwent pairwise retesting.

We fully agree with the Reviewer that comparing data from the same lab addresses a notion of reproducibility. Still, if our method finds that two networks agree with each other, it means that there is not much room left for reproducibility (or other) issues. Note that the nature of these datasets is subtle. Although the search space of these three networks overlaps to a large extent, none of the interaction sets have been simply included in later editions of the data. Not only the methodology has been changed in later experiments, but the new experiments have been performed in a systematic way, without any influence from previous experiments within or outside of the same lab. As a result, only 25% of PPIs in HI-I-05 are present in HI-II-14, while only 40% of the PPIs in HI-II-14 are present in HuRI. Therefore, it is of interest to compare these networks to understand the possible origin of these differences.

1.6) Figure 4d HURI is not connected, why is that the matrix shows strong connections. Why is Venkatesan a central node in the graph?

Figure 4c and 4d are designed to show different aspects of the agreement network. In Figure 4c we show the Normlap score without the standard deviation while in 4d we connect the networks only if the observed overlap is not statistically lower than the positive benchmark, which takes the standard deviation into account. Although HuRI has high Normlap scores compared to some networks, we have statistical power to see that the observed overlap is statistically different from the positive benchmark and thus not connected to any other networks. Venkatesan is an experimental dataset constructed from four Y2H assays using a stringent version of the Y2H method. Potential reasons that make Venkatesan compatible with most datasets include but are not limited to:

- Venkatesan only includes high-confidence PPIs, supported by at least two Y2H reporter assays.
- Venkatesan is a small dataset, including only 195 proteins, reducing the statistical power of finding significant differences.

A discussion of this point is now included in the manuscript in lines 263-267

Reviewer #2:

2.1) The authors present a computational framework to assess overlap among networks. While the approach is conceptually interesting and innovative, it is not clear that the calculations are suited for the claims. Also, the discussion of problem and other approaches, although informed, is not entirely up to the state of the art in the field. This is exemplified by the first sentence of the abstract stating that 'significant overlap' with a reference network was 'current best practice'. As the manuscript is largely focused on protein interactions, this is not correct. Current best practice is to test a random subset from the new dataset in a benchmarked orthogonal validation assay.

We agree with the Reviewer that in a broad sense, the current 'best practice' includes additional experiments in a validation assay. However, strictly on the computational side, there is only the naive overlap available as the current best practice. We have updated this sentence to clarify this point in lines 7-8. Even if additional experiments are performed, what happens is still a network comparison. A validation assay gives yes/no answer to each link depending on whether they have been found in the validation assay. This means that *de facto* a network overlap is calculated between the (typically mostly

uncharted) network of the validation assay and the tested network. Even though the networks of the frequently used validation assays have not been fully constructed, for example there is no MAPPIT or GPCA network yet, there is conceptually an underlying network comparison. And as such it falls into the problem that our computational work is meant to improve upon - naive results being biased by degree inconsistencies and the lack of a positive statistical benchmark.

2.2) Importantly, as the authors point out, there is no inherent measure to dataset precision, and false positives in both networks are handled as 'true positives'. At the end, the authors make a revealing point by suggesting that their normlap score depends on noise (i.e. poor quality) and that reliable assessment of noise (dataset quality) requires other approaches. Thus, even in the manuscript the authors agree that their approach is not suited for quality control as insinuated in the abstract and throughout the text.

As we have shown in our work, taking degree inconsistency into account in any network comparison is an important step. We also made clear that there can be additional factors to take into account, depending on the purpose of network comparison. Aiming at quality control is just one of the applications that done properly might require more steps than degree control. However, in all those cases where we see no statistically significant deviation from a perfect network overlap, we can conclude that there is no sign of quality issues, see for example I3D and Ito-core. However, when degree inconsistency is clearly not the only reason for discrepancy, additional work is needed to identify the factors at play. For example, explicitly quantifying the amount of noise in a dataset requires a noise model in general, which goes beyond the scope of this manuscript. Even in such cases, the Normlap score is still more indicative of the quality of networks compared to the observed raw overlap, as demonstrated by Figure 3b, S1c. To make these points more clear, we improved the manuscript in lines 348-352.

2.3) How do the authors handle search space issues (see Venkatesan et al, 2009): if we have two perfect representations of biology but for non-overlapping sets of 200 genes, each, there will be no overlap, thus suggesting, to anyone thinking that overlap had any relation to data quality, that this must be low. It is not clear how this scenario was handled in the process of forming a union and subsequently splitting it again.

We agree with the Reviewer that in the case when two networks cover distinct genes and have no overlap, we cannot draw any conclusions as the overlap will always remain zero, even if constrained by the degree sequence. We made this more clear in our text (lines 506-509) and Figure 4.

2.4) Ln 29: yes, no assay can capture all interactions but this is a statement about a single biological modality, when switching biological modalities, this is a different statement, which is at the same time trivial and not an 'example' of the first statement.

We thank the Reviewer for catching this inconsistency. We have rearranged and integrated the text to make the flow of the logic more clear (see lines 26-29).

2.5) Ln 30. PCA is an assay that interrogates 2 interaction partners, but is not a 'binary' assay in a sense of detecting predominantly direct interactions. In fact, I believe the QC data in the manuscript suggest the data are closer to co-complex datasets - please double check.

We refer to PCA as a binary method as it identifies interactions between pairs of proteins. We concur with the Reviewer's observation that PCA does not necessarily indicate predominantly direct interactions. We are grateful to the Reviewer for bringing this to our attention and changed the manuscript in lines 29-31, 243-246 to avoid confusion.

2.6) Ln 44: no, the question if overlap suggests data quality has been addressed and it is not unless many aspects of the dataset are taken into account – including completeness and precision. The authors fully omit discussion of the interactome mapping framework (Venkatesan et al. Nat Meth 2009), and prior work on the yeast interactome Yu et al, Science 2008, and QC approaches based on confidence scoring as a state-of-the-art for quality control (Braun et al, Nat Meth 2009).

We fully agree with the Reviewer that the raw overlap does not suggest data quality unless many aspects are taken into account, one of those factors being degree inconsistency, which we address computationally for the first time here, see also our response to point 2.2. We have now included the discussion of Venkatesan et al.'s framework in lines 48-53. We are aware that experimentally mapping the interactome with proper quality control is a subtle and demanding challenge. However, discussing steps and issues with specific interactome mapping methodologies goes beyond the focus of our study. Our aim is to improve on the routinely applied computational step of comparing published datasets, produced by a broad range of methodologies.

2.7) Ln 64: why would one expect or need a consistent degree sequence – the authors discuss completely different biological modalities.

We agree with the Reviewer's point that there is no reason to expect consistent degree sequences between different modalities. This is part of the reason why comparing networks of different modalities necessitates the use of our methodology since the degree sequences are very different between different modalities. We have updated the text in lines 61-65 accordingly.

2.8) Ln 88: Please justify why I3D is taken as 'gold standard'. Due to its nature as being supported by structures one would expect this to be similarly biased and incomplete, no? Please rephrase and justify the use as a reference (which I strongly suggest as alternative to 'gold standard') despite such problems.

We thank the Reviewer for their valuable advice and would welcome any further suggestions for alternative terminology to replace "gold standard" or "reference network" in this manuscript. We initially thought of using the term "reference network" as an alternative to "gold standard." However, the fact that HuRI and YeRI are named as "reference networks" may lead to confusion. In line with the Reviewer's suggestion, we have now changed "gold standard" to "reference network" throughout the manuscript. Additionally, we have provided further clarification on the meaning of "reference network" in this context, stressing that it does not imply a perfect or unbiased network. We have also included motivations to use I3D as a reference network as an illustration, see lines 94-98.

2.9) Ln 102: 'small overlap' between networks of different biological modalities (GI, PPI) could suggest that 'most of these networks are problematic' – this phrase is aggravating. How could any person expect a large overlap between networks in different biological modalities and even make a strong assumption to this extent to then conclude from the lack thereof, that this is reflective of a poor data quality. I do

agree with the authors that assessment of overlap can be useful or interesting, but phrasing an overlap analysis as a way to assess quality of one or the other compared networks even across biological modalities is, my apologies, ignorant.

We apologize for the misstatement. We have revised the text to clarify the logic, which can be seen in lines 109-116. We did not mean to suggest that low overlap between different modalities implies low quality. Rather, our point is that low overlap between the same modalities may raise questions about quality, while low overlap between different modalities could primarily indicate differing biological content and mechanisms.

2.10) Ln 129: It is not fully clear what the authors are doing. Forming the union of two networks is understood, but it is unclear how a split of this union into two networks yields any overlap. How do the authors handle double edges, are they kept as two or collapsed into one? Please specify the approach. Also, conceptually, it is not clear how the approach to mix and split is justified. Partly this results from the confusion as to how the overlap among two partitions of one union network yields any overlap. But also, it is not clear, how mixing of networks from different biological modalities, or of binary and co-complex interactions is justified and meaningful.

In line with the concept of taking the union, we have consolidated the double edges into a single edge in order to capture the overall structure of the network. To provide greater clarity, we have added further explanation of the methodology in lines 140-143. We have also replaced the word "split" with "sample." The two samples are not partitions, as each follows the original degree sequence from the same union. Therefore, as shown in all of our examples, the overlap between the samples from the union serves as an upper bound for what could be observed for the input networks. Please also see our response to point 3.2 for the reasons behind comparisons of networks across different modalities.

2.11) Ln 130: please be specific, which complete graph is available?

The complete graph refers to a fully connected graph consisting of all nodes in a given network. To make this clear, we updated the text in lines 149-150.

Ln 181 - 183: yes, understanding different modalities is important, but a degree inconsistency does not hinder further interpretation but reflects difference centrality of a protein in different biological modalities – a TF regulating thousands of genes, is not expected to also be a central hub in the metabolic network. So, different degree distributions do NOT preclude further interpretation.

We thank the Reviewer for this note. We have revised the word "hinder" as it was indeed too strong and did not accurately convey our intended meaning. What we meant was that when interpreting the results, degree inconsistency gets mixed into the interplay of other confounding factors, making it hard to decipher if the observed differences are due to any additional biases, issues, or even biological differences. By taking node degrees into account as a confounding factor, we can now see if there is anything else at play.

2.12) 185: the genetic network is not mapping the wiring diagram – please rephrase

Our understanding is that the authors of the dataset agree with our statement, as also indicated by the title of the reference: Costanzo, M. et al. A global genetic interaction network maps a wiring diagram of cellular function. Science 353 (2016).

2.13) Ln 216 and following: what exactly do you mean to express by ‘compatible’ in biological terms.

Our definition of compatibility is in the statistical sense (see lines 161-166). Relating compatibility to the underlying biology requires case-by-case evaluations, as there could be multiple things behind compatibility. In our definition, if two networks of the same modality are compatible it means there is no room left for quality issues, such as in the case of I3D and Ito-core. If two networks of different modality are compatible, such as YeRI and co-localization, it implies that as much as the degree allows, the YeRI PPIs are always supported by co-localization pairs and vice versa.

2.14) Ln 228: structurally supported interactions could have informed functional annotations, or merely act in the same processes, e.g. SH2 domains, SH3 domains, death domains. Please keep this circularity in mind in the discussion.

We agree with the Reviewer and have now added additional information to the manuscript to mention this circularity as indicated in lines 253-256, stating that the high scores between PPI networks and the co-annotation network could potentially be a result of circularity.

2.15) Ln 257: No, dataset quality is NOT assessed by testing promising links, but a random sample. This is critical.

We apologize for the confusion and have rephrased this part, please see lines 296-297.

2.16) Ln 263: the ‘degree inconsistency’ is a consequence of different biochemical biases of different assays, and this is captured by the benchmarking. Therefore the approach is the only way to assess data quality, and better so than any overlap. Please check the relevant literature and rephrase (Yu, 2008, Venkatesan 2009, Braun 2009).

We fully agree with the Reviewer that experimental techniques are *in principle* superior compared to solely computational approaches to assess data quality. However, experimental methods are also limited in their scope due to the often high cost. As there is currently no MAPPIT or GPCA network, the real degree of nodes in these assays is unknown. Given this, it is a meaningful task to improve upon existing scalable computational approaches. Please see also our response to 2.1. We would like to emphasize again that our framework is not only an important step towards this goal, but has broader applications, beyond quality control. We have revised our text accordingly in multiple points in our text (lines 9-11, 164-166) to make it clear.

Reviewer #3:

The article “a positive statistical benchmark to assess network agreement” written by Hao and Kovacs introduced a new generative positive benchmark approach and associated evaluation metric score (Normlap score) for quantitatively assess the network quality. The paper is well written with enough conceptual description and examples. The idea of mixing network and re-sampling is conceptually simple

but neat. I have the following questions regarding this paper, and hope they could be addressed before publication.

3.1) In the section of “Agreement networks of networks”, the little agreement between HuRI and BioPlex is surprising but maybe expected. Many papers claimed that yeast-hybrid approach in general has a very high false positive rate as the hybridizing reaction happens in yeast nucleus that is not the original cellular condition. Could authors add one section to benchmark the two networks using I3D-H, I am very curious to see the evaluation result between HuRI/BioPlex with I3D-H using the new metric.

This is a great point! We have compared I3D-H with HuRI and BioPlex in Figure 4c. It shows that neither HuRI nor BioPlex is compatible with I3D-H. However, HuRI vs. I3D-H shows a higher score compared to BioPlex vs. I3D-H, indicating that the HuRI is closer to I3D-H. We agree that it would be interesting to see how I3D-H validates HuRI/BioPlex like what we did in Figure 5. Unfortunately, the published version of HuRI/BioPlex does not come with a ‘quality score’ for each link so we are not able to threshold the datasets accordingly.

3.2) In the section of “Threshold and validate experimental assays”, the authors tried to validate yeast GI-PSN to I3D and co-complex networks. To me, this doesn’t make sense, GU-PSN is genetic interaction network. Genetics interaction is very complicated itself, where the two genes might function in parallel pathways, in same complex/pathway, in cross-talked pathways...To compare genetics network with physical pathways are like comparing apple to orange. In the section of “Normalized network overlap”, the authors again discussed using Normlap score to compare GI network with four PPI network and co-complex network. First, I don’t think the comparison is meaningful as genetic network and physical network are apple and orange.

We fully agree with the Reviewer that GIs are a world of difference compared to PPIs. However, this does not make such comparisons meaningless. For example, both the GI and GI-PSN networks are routinely compared with PPI networks and networks of other modalities (c.f. Fig. 6C and S16 B,D in Costanzo 2016), especially for the reason that the Reviewer mentioned: to learn about the underlying biology behind those complicated genetic interactions. We have cited the main reasons and conclusions of comparing GI to PPI networks in lines 205-216 and showed that we reached an improved yet consistent picture. For example, negative genetic interactions between essential genes show a significant overlap/compatibility with PPIs and co-complex networks, unveiling that a cell is less tolerant to a joint loss of function mutation (negative GI) in the same essential pathway or complex. The opposite direction is also meaningful, i.e. comparing PPI networks to GI or GI-PSN networks, see for example the Extended Data Figure 4i in the YeRI study (ref 5). In human networks, although there is no systematic GI network, PPI and PSN networks are often compared to different modalities like co-expression networks (for example Extended Data Fig. 5 in Luck 2020, the HuRI paper). The reason behind such comparisons is to learn about functional relationships between the observed PPIs.

3.3) Second, they claimed overlap fraction of GI_NxN_neg is much smaller than GI_ExE_neg. I don’t see how the claim draws any meaningful biological conclusion, as the size of GI_NxN_neg is much larger than GI_ExE_neg.

The overlap fraction is defined as the observed overlap divided by the size of the **smaller** of the two comparing networks, in this case, the size of I3D. Although the size of GI_NxN_neg is much larger than GI_ExE_neg, it still has less overlap with I3D. This provides further support for our conclusion that GI_ExE_neg is more compatible with PPI networks than GI_NxN_neg, which means that mutations in two genes that correspond to a PPI are more likely to lead to reduced cell fitness (negative GI).

3.4) One conceptual concern is the biological contexts/cellular contexts are not discussed in this paper. For example, networks mapped from different cell lines/conditions...are of course different. Could the authors make some comments about this and their newly developed approach?

This is an excellent question, and we fully agree with the Reviewer that biological context can have a profound impact on the measured networks. Part of the observed inconsistency in Fig. 4 may originate from the fact that they include PPIs detected in different cellular contexts, as an aspect of biophysical differences. Addressing this point, we have included the analysis of an additional dataset, see Fig. S4, that compares human PPI networks derived from different cells and we discuss the results in lines 268-274. This comparison also relates to point 3.1, namely why HuRI is not compatible with most networks. One of the reasons could be attributed to differences in biological contexts.

3.5) The idea introduced in this paper is theoretically sound and interesting with many abstract/conceptual examples. But I don't see a clear way to use it for facilitating new biology discovery, which I think this is the final goal of biological network mapping. Thus, I think the authors should add a section to discuss/demonstrate how to use the new statistical framework to help new biology discovery. We thank the Reviewer for the positive evaluation of our work and especially for the constructive suggestion. We believe that comparing networks across different modalities already facilitates new biology discovery, as discussed in point 3.2. Additionally, we have added a new section to the manuscript (lines 353-372) to discuss relating our statistical framework to new biology discovery. We believe that our results will provide a better entry point to such an endeavor and are excited to work with our collaborators on improving upon existing experimental pipelines using our methodology.

3.6) There are a lot of p-values listed in this paper. The names of statistical tests are usually missing and I don't have a clear idea how these p-values are computed. The authors should clarify them in the revised version.

We have added the explanation of how the p-values are calculated in the Methods - "Construction of positive and negative benchmarks" section, please see lines 496-502.

3.7) In the abstract, the authors mentioned this is a "maximum entropy framework", which is confusing. I don't see the connection between the approach introduced in this paper and information theory. The authors should clarify this in the revised version.

We thank the Reviewer for bringing this point up, which is a key step in our methodology. We utilize the maximum entropy framework to create "random" networks. In fact, the very definition of a "random" network here means the network ensemble with a maximum entropy that complies with a set of constraints. While there are alternative approaches to create a negative benchmark, such a maximum entropy framework is currently the only viable way to construct a positive benchmark for network

overlap, as alternative definitions of random networks cannot efficiently deal with the required combination and large number of soft and hard constraints. We have included additional clarification in the revised manuscript, see lines 476-479.

3.8) The colors used in this paper are bad. It is hard for me distinguish lines in plots (such as Figure 3, 5, S2). The authors should print the paper and see how difficult it is for readers to read these plots. So, colors should be changed and lines should be made thicker in all plots.

Thank you for your feedback. We appreciate your suggestions and have made changes to the colors and line thickness in all plots accordingly.

REVIEWERS' COMMENTS

Reviewer #1 (Remarks to the Author):

The authors addressed the comments and point for consideration through large text revisions. In many cases these changes are real improvements facilitating a better understanding of the rational, approach and outcomes of the study. I suggest to go ahead with the manuscript.

Reviewer #2 (Remarks to the Author):

The manuscript is greatly improved in the scope of explanations and the depth of the discussion.

Just a few minor comments:

please rephrase first abstract sentence to: “[...] current computational best practice [...]”.

Ln 37: knowing the overlap among networks of different modalities seems to be more helpful than essential. perhaps tone down.

Ln 41: the experimental quality assessment and the accompanying interactome framework do explain well the overlap – it is a consequence of search space, and both assay and sampling limitations. Please rephrase the motivation accordingly. The normlap scores is still very helpful and perhaps complementary.

Lines 45 and following: this is a misrepresentation of the situation. Given the benchmarking data in Luck et al., it is clear that the low overlap between assay versions is NOT due to poor network quality. In fact, the assays were selection with the purpose of having complementary (non-overlapping) detection profiles and thereby provide more coverage. While the general point is correct and important, the paragraph as it is now seems to misrepresent the previous work.

Reviewer #3 (Remarks to the Author):

I think the authors did a good job to address my comments.

The only remaining issue is 3.5, I think the newly added discussion doesn't provide much meat as it is too abstract. I was hoping the authors could pick a detailed biology example, like a novel gene discovery, a new complex function prediction etc...I think what the biology community really wants to see is they can use this tool/metric/approach to solve some down-to-earth biology problem.

Apart from 3.5, the authors addressed all my other questions.

REVIEWERS' COMMENTS

Reviewer #1 (Remarks to the Author):

1.1) The authors addressed the comments and point for consideration through large text revisions. In many cases these changes are real improvements facilitating a better understanding of the rationale, approach and outcomes of the study. I suggest to go ahead with the manuscript.

Thank you for your positive feedback and your recommendation to proceed with the manuscript. We appreciate your time and effort in reviewing our work and providing constructive feedback.

Reviewer #2 (Remarks to the Author):

The manuscript is greatly improved in the scope of explanations and the depth of the discussion. Just a few minor comments:

2.1) please rephrase first abstract sentence to: “[...] current computational best practice [...]”.

Thank you for your suggestion. We have revised the manuscript accordingly.

2.2) Ln 37: knowing the overlap among networks of different modalities seems to be more helpful than essential. perhaps tone down.

Thank you for your feedback. We have toned down our language in the revised manuscript.

2.3) Ln 41: the experimental quality assessment and the accompanying interactome framework do explain well the overlap – it is a consequence of search space, and both assay and sampling limitations. Please rephrase the motivation accordingly. The overlap scores is still very helpful and perhaps complementary.

Thank you for the comment. We have revised the text accordingly.

2.4) Lines 45 and following: this is a misrepresentation of the situation. Given the benchmarking data in Luck et al., it is clear that the low overlap between assay versions is NOT due to poor network quality. In fact, the assays were selected with the purpose of having complementary (non-overlapping) detection profiles and thereby provide more coverage. While the general point is correct and important, the paragraph as it is now seems to misrepresent the previous work.

Thank you for bringing this to our attention. We agree with the reviewer that the low overlap between HuRI assays is not due to poor quality. We have revised the related text to make the point more clear.

Reviewer #3 (Remarks to the Author):

3.1) I think the authors did a good job to address my comments. The only remaining issue is 3.5, I think the newly added discussion doesn't provide much meat as it is too abstract. I was hoping the authors could pick a detailed biology example, like a novel gene discovery, a new complex function prediction etc... I think what the biology community really wants to see is they can use this tool/metric/approach to solve some down-to-earth biology problem. Apart from 3.5, the authors addressed all my other questions.

Thank you for your feedback. We appreciate your positive comments and are glad to hear that we were able to address your concerns. Regarding 3.5, we acknowledge your point and agree that a concrete example would be beneficial to demonstrate the utility of our approach in solving real-world biology problems. We have already demonstrated several applications of our method and outlined potential directions for new biological discoveries. Applying the method to facilitate the discovery of specific new complex functions requires additional work in integrating relevant datasets and exploring underlying biological mechanisms that are beyond the scope of this study.

As a follow-up project, we are currently exploring improved functional predictions for NF- κ B signaling, an extension of our earlier computational and experimental work in Ref. [5]. In the meantime, we welcome the community to apply our methods and believe that our methods will help facilitate new biological discoveries like complex function predictions, etc.